# Toxic Effects of Paclobutrazol on Developing Organs at Different Exposure Times in Zebrafish

**DOI:** 10.3390/toxics7040062

**Published:** 2019-12-06

**Authors:** Wen-Der Wang, Chang-Yi Wu, Bonick Kartini Lonameo

**Affiliations:** 1Department of Bioagricultural Sciences, National Chiayi University, Chiayi City 60004, Taiwan; bonick_kartini@yahoo.com; 2Department of Biological Sciences, National Sun Yat-sen University, Kaohsiung 80424, Taiwan; cywu@mail.nsysu.edu.tw

**Keywords:** Paclobutrazol, toxicity, Zebrafish, exposure timing

## Abstract

To enhance crop productivity and economic profit, farmers often use pesticides that modulate plant growth and prevent disease. However, contamination of ecosystems with agricultural pesticides may impair the health of resident biota. Paclobutrazol (PBZ), an aromatic-containing triazole, is widely applied to many crops in order to promote flowering and fruit setting, while also regulating plant growth and preventing fungus-related diseases. Due to its high mobility, high stability and potential for bioaccumulation, the risks of PBZ to the health of organisms and ecological systems have become a serious concern. In previous studies, we documented the toxicity of PBZ on developing heart, eyes, liver, pancreas and intestine of zebrafish. In this study, we sought to further understand the developmental stage-specific impacts of PBZ on digestive organs and other tissues. Zebrafish were exposed to PBZ beginning at different embryonic stages, and the toxic effects on organs were evaluated at 120 hpf (hours post-fertilization) by in situ hybridization staining with tissue-specific marker genes, such as liver, intestine and pancreas. Unsurprisingly, early-stage embryos exhibited higher sensitivity to PBZ-induced death and developmental hypoplasia of digestive organs. Interestingly, the developing liver and pancreas were more sensitive to PBZ than intestine when embryos were exposed at early stages, but these tissues showed lower sensitivity at later stages. Our delineation of the differential toxic effects of PBZ on developing organs at different exposure timings can serve as a powerful reference for further studies into the mechanisms of PBZ organ toxicity.

## 1. Introduction

Crop productivity and economic profit may be improved by the application of pesticides. As such, these xenobiotics are widely used by farmers to promote growth and prevent disease in plants. However, contamination of ecosystems with agricultural pesticides can have severe negative impacts on environmental and human health. Paclobutrazol (PBZ) is an aromatic-containing triazole chemical that inhibits gibberellin biosynthesis in plants. Gibberellins (a class of diterpenoid phytohormones) regulate plant growth and development, including stem elongation, germination, seed dormancy, flowering, and senescence in fruit and leaves. At least 136 types of gibberellins have been identified, but only a few appear to be bioactive, including GA1, GA3, GA4, and GA7 [1]. In plant gibberellin biosynthesis, ent-kaurene oxidase is a key enzyme that converts the intermediate metabolite, ent-kaurene, to ent-kaurenoic acid, which is then further processed by other enzymes to produce various gibberellins. Since PBZ efficiently inhibits ent-kaurene oxidase activity to block gibberellin biosynthesis in plants, it is widely applicable and has been used on flowers, fruits, vegetables and other crops to promote flowering and fruit setting, while also regulating plant growth and preventing fungus-related diseases. These wide-ranging functions of PBZ have caused it to become one of the most widely used pesticides in the world, and the use of PBZ has increased every year since its first approval in 1985. There are two different methods of application for PBZ: solution-spraying on the leaves/stem and powder-burying to expose the roots. In real-world farming applications, PBZ is often used in foliar sprays at concentrations ranging from hundreds to thousands ppm; doses are targeted at 4–202 grams per individual, depending on the size and species of the plant (Massachusetts Department of Agricultural Resources). PBZ has a half-life of 43 to 618 days (average 182 days) in soil under aerobic conditions; the actual half-life depends upon the physical and biological conditions, including the surrounding temperature, tightness of soil and bacterial contents. In surface water, the half-life of PBZ is 164 days. Thus, PBZ exhibits high chemical stability, and its decomposition is largely unaffected by acidic, basic or neutral conditions. The application of PBZ on agricultural land will eventually lead to contamination of water systems due to rain and water washing. In the natural environment, the high mobility and high stability of the molecule is potentially a serious concern; PBZ concentrations of 4.2 μg/L have been reported in groundwater, and 150 mg/kg was found in soil at soil-injected sites [2,3]. Surprisingly, the PBZ content in seawater from the Jiulong Estuary of China and the West Sea of Xiamen was found to be as high as 119.6 ng/L [4].

Damalas and Elefterohorinos reported that workers who apply pesticides and people who live near agricultural fields are the most at risk to exposure of agricultural pesticides [5]. Rasmussen et al. further argued that periodic contamination with agricultural pesticides may especially impair the structure and/or function of stream biota, and agricultural streams are among the most impacted systems on Earth [6]. Indeed, the co-occurrence of numerous pesticides in agricultural streams has been reported frequently in scientific studies and monitoring programs.

Soil treatment (1 mg/L) and foliar treatment (25 mg/L) with PBZ can improve the photosynthetic activity and water balance of tomato plants (cv. Precador). PBZ treatment has also been shown to accelerate fruit formation and increase early fruit yield. Treatment of plants with PBZ at the time of pricking out ensures the fruits will not carry any residual retardant, mitigating the potential harm to human health [7]. However, many lines of evidence indicate that the misuse of PBZ leads to contamination of both the environment and the food supply, posing risks to the health of humans and animals [8,9,10,11,12].

In order to explore the potential toxicity of PBZ in vertebrate organisms, we examined its toxic effects in a zebrafish model. Zebrafish is a small freshwater teleost that is transparent, fast growing and exhibits highly conserved developmental programs. Moreover, the housing and husbandry space required for this species is small and low cost. In addition, zebrafish have high fecundity (each breeding pair can produce 300 embryos per week). Based on these major advantages, zebrafish have become widely used and are now a globally accepted model in toxicological research, with specific approval from the U.S. Food and Drug Administration for new drug discovery research. Our previous study revealed that PBZ disrupts the development of heart and craniofacial cartilage in zebrafish embryos, and it decreases survival and hatching rates. PBZ specifically affects cardiac looping, which may lead to a slower heart beat accompanied by pericardia edema. The pesticide also decreases the population of migratory neural crest cells, which give rise to craniofacial cartilage [13]. We also reported that PBZ has toxic effects on embryonic development of digestive organs and retinal photoreceptor cells in zebrafish [14]. PBZ exposure leads to developmental hypoplasia of digestive organs, including liver, intestine and pancreas, which are derived from the same embryonic endoderm. Similar to other aromatic-containing chemicals, PBZ weakly induces *cytochrome p450 1a1* (*cyp1a1*) by activating AhR signaling, and our previous study implicated AhR activity as an important mediator of PBZ toxicity to digestive organs [15]. In addition, retinal photoreceptor cell development requires retinoic acid-related signaling, and we previously found that the expression of *aldehyde dehydrogenase 1a2* and *1a3* (*aldh1a2* and *aldh1a3*), key enzymes for retinoic acid biosynthesis, were reduced in PBZ-treated embryos; importantly, PBZ-impaired development of retinal photoreceptor cells could be rescued by ectopic retinoic acid. Thus, in addition to AhR, retinoic acid signaling is also involved in the mechanisms underlying PBZ embryonic toxicity [14]. 

Although PBZ toxicity has been studied for several years, the molecular basis of its effects are still not clear. In this study, we exposed zebrafish embryos to PBZ at different stages and concentrations; we then examined the effects of PBZ exposure on survival rate, percardium, head skeleton, and digestive organ development in zebrafish at 120 hpf. We found that embryos exhibit dose-dependent toxicity phenotypes, consistent with the results from our previous studies. Furthermore, more severe embryonic developmental toxicity was observed when embryos were exposed to PBZ at earlier stages. Surprisingly, zebrafish embryos exhibited defective head skeleton formation after 3.4 and 17 μM PBZ treatment when the initial exposure time was 48 and 60 hpf (hours post fertilization); this timing suggests that PBZ not only reduces neural crest cells, but it also impairs the cartilage differentiation process. While all digestive organs (pancreas, intestine and liver) are derived from embryonic endoderm, embryos exposed to PBZ at different stages exhibited different severities of effects in each tissue. Overall, our study reveals the developmental toxicity of PBZ at different concentrations and initial exposure stages, providing a powerful reference for future mechanistic studies on the toxic effects of PBZ in developing heart, head skeleton and digestive organs.

## 2. Materials and Methods 

### 2.1. Ethics Statement

The study plan for use of zebrafish, animal care and all experiments were conducted in compliance with the Organization for Economic Cooperation and Development (OECD) [16] and all protocols were approved by the Institutional Animal Care and Use Committee (IACUC) at National Chiayi University (Animal Use Protocol #104043, approved 15 January 2016). All studies were carried out in strict accordance with the IACUC guidelines.

### 2.2. Chemicals

Paclobutrazol (PBZ; α-tert-Butyl-β-(4-chlorobenzyl)-1H-1,2,4-triazole-1-ethanol) was purchased from Sigma–Aldrich (Cat No.46046). A stock solution of PBZ was prepared in dimethyl sulfoxide (DMSO) at a concentration of 1.02 M and stored at −20 °C.

### 2.3. Fish Maintenance, Embryo Treatment and Collection

The AB zebrafish strain (*Danio rerio*) was used. Zebrafish were maintained under standard laboratory conditions at 28.5 °C under a day-night cycle (14 h light/10 h dark) with an automatic timer. Zebrafish embryos were spawned by natural mating, and at the 1- to 8-cell stage, embryos were collected and evaluated for quality. Qualified embryos were cultured in sea salt egg water (0.0375% sea salt in deionized distilled water) containing various concentrations of PBZ [0.34 μM, 3.4 μM, 17 μM] or DMSO (0.1%, *v*/*v*) beginning at 24, 36, 48, 60, 72, or 96 hpf, and incubated at 28.5 °C. The survival ratios and organ development defects were determined at 5 dpf (days post fertilization). Culture media was not renewed during experiments. For some experiments, 1-phenyl-2-thiourea (final concentration of 0.2 mM; Sigma–Aldrich) (Sigma–Aldrich, Louis, MO, USA) was supplied to the media at 20–22 hpf to inhibit pigmentation, as described previously [17]. Embryos were fixed at 5 dpf in 4% paraformaldehyde (PFA) (PBT; 1× PBS/0.1% Tween-20) at 4 °C overnight, followed by dehydration and storage in 100% methanol at −20°C prior to use in experiments.

### 2.4. Survival Rate of Embryos

Zebrafish embryos (100 embryos/plate) were exposed to different concentrations of PBZ (0.34, 3.4, 17 μM) or 0.1% DMSO and incubated at 28.5 °C. The dead embryos were removed daily and recorded. Quadruplicate independent experiments were performed to evaluate the survival rate.

### 2.5. Whole Mount in Situ Hybridization

Whole mount in situ hybridization was performed as described previously [18]. Digoxigenin (DIG)-labeled antisense riboprobes (*trypsin*, *lfabp*, and *ifabp*) were synthesized by in vitro transcription reactions. DIG-labeled riboprobes were detected using alkaline phosphatase-conjugated anti-DIG antibodies; the hybridization signals were developed using NBT/INT solution as a substrate (Roche). Embryos were imaged by using Olympus microscope imaging system.

### 2.6. Cartilage Staining 

Embryos at 5 dpf were anesthetized and fixed in 4% PFA at 4 °C overnight. Embryos were washed with PBT several times and bleached in 1 mL of 10% H_2_O_2_ supplemented with 100 μM KOH for 1 h at room temperature. The samples were then stained in 0.1% Alcian Blue dissolved in acid ethanol (75% ethanol, 1% concentrated hydrochloric acid) at room temperature overnight. Stained embryos were washed extensively in acidic ethanol, dehydrated and stored in 80% glycerol. The staining was imaged using an Olympus microscope.

### 2.7. Statistical Analysis

The effects of PBZ treatment on zebrafish embryonic survival rate and pericardial edema were determined based on the ratio of surviving embryos to total treated embryos and the ratio of embryos with percardiac edema to surviving embryos, respectively. The toxicity of PBZ to head skeleton, pancreas, liver, and intestine was determined based on the frequency of the observed abnormalities. Data are presented as the mean ± standard deviation, and one-way Analysis of variance (ANOVA) followed by Fisher’s least significance difference test was performed to analyze the significance of differences. All statistical analyses were performed using SPSS 21.0 (IBM, Armonk, NY, USA).

## 3. Results

### 3.1. Embryonic Stage of Initial PBZ Exposure Influences Survival Rate of Zebrafish 

Zebrafish embryos were exposed to vehicle control (0.1% DMSO), or 0.34, 3.4, 17 μM PBZ beginning at different embryonic stages (24, 36, 48, 60, 72, or 96 hpf). The survival ratios were then determined at 5 dpf. Embryonic survival rate was dramatically reduced in 17 μM PBZ-treated embryos compared to lower concentrations in the same treatment-stage cluster. A dramatically reduced survival rate was observed when zebrafish were exposed to high-concentration PBZ (17 μM) beginning at early embryonic stages (24 and 36 hpf), as compared to embryos exposed to 17 μM PBZ beginning at later embryonic stages (48 to 96 hpf). For embryos exposed to PBZ at 48 hpf, the survival rate was not significantly reduced by 0.34 μM PBZ treatment compared to control, but it was reduced in 3.4 and 17 μM-treated embryos. For embryos first exposed at 60, 72, and 96 hpf, the survival rate was not reduced in 0.34 or 3.4 μM groups, but it was significantly reduced by 17 μM treatment (Figure 1). These results indicate that exposure of embryos to PBZ beginning at early stages (younger than 48 hpf) increases fish mortality compared to exposures beginning at later embryonic stages.

### 3.2. PBZ Dose-Dependently Induces Pericardial Edema after Early-Stage Exposure

The heart is critical for blood circulation and transport of nutrients in the body. Perturbation (genetic or environmental) of cardiac development can easily lead to catastrophic heart defects and subsequent embryonic/fetal demise. Moreover, congenital heart diseases, which include both structural and functional defects, occur in about 5% of live births. In our previous study, we noted pericardial edema developed after treatment of embryos with high-concentration PBZ (~68μM). Pericardial edema always accompanies heart tube elongation, heartbeat malformation and cardiovascular dysfunction. To determine the initial exposure stage that results in obvious pericardial edema, we examined the hearts of 5 dpf fish treated with PBZ at the concentrations of 0 (0.1% DMSO), 0.34, 3.4, or 17 μM beginning at different initial exposure stages (Figure 2). Our results showed that similar percentages of embryos exhibited pericardial edema at 120 hpf when embryos were first exposed to PBZ at 24 and 48 hpf, but this phenotype was dramatically reduced when exposure was initiated after 48 hpf, indicating that heart development of later staged embryos shows increased tolerance to PBZ. Interestingly, a few embryos still exhibited pericardial edema at 120 hpf after PBZ exposure beginning at 60 or 72 hpf. At these stages, the heart tube has already looped and matured, suggesting PBZ may still exhibit some toxicity to the matured heart.

### 3.3. PBZ Impairs Head Skeleton Development at Early Embryonic Stages and Also Disrupts Precursor Cell Differentiation

The vertebrate head skeleton consists of ventral pharyngeal arches and dorsal neurocranium, which are mostly derived from the cranial neural crest. This hard structure supports the features of the face and forms a cavity for the brain. Many studies have demonstrated that malformations in the fetal head skeleton can be induced by exposure to toxicants during embryonic stages [19,20,21,22]. Zebrafish develop a simple pattern of early larval cartilages and bones, which is highly conserved among vertebrates [23,24]. Our previous study demonstrated that exposing zebrafish embryos to PBZ at 2 hpf dose-dependently resulted in head skeletal malformations [13]. Here we further examined which embryonic stage is the critical point at which PBZ exposure can induce malformations of the head skeleton. To address this issue, we treated different embryonic stages with multiple concentrations of PBZ and analyzed head skeleton development by alcian blue staining. Control and PBZ-treated embryos at 5 dpf were grouped into categories of ‘normal’, ‘mild’, or ‘severe’ defects based on the alcian blue-stained morphology of head skeleton (Figure 3A). A statistical comparison indicated that embryos initially exposed to PBZ at 24 and 36 hpf exhibited similar rates of mild and severe malformation of head skeleton in 3.4 and 17 μM PBZ (3.4 μM-treated embryos: 41.4 ± 1.2% and 38.3 ± 2.3% mild effects at 24 and 36 hpf, respectively; 17 μM-treated embryos: 35.3 ± 2.7% and 35.7 ± 3.5% mild defects, 63.2 ± 1.5% and 63 ± 2.4% severe defects at 24 and 36 hpf, respectively). Interestingly, less than 40% of embryos exposed at 60 hpf presented mild defects in head skeleton, and no defects were observed in head skeleton when embryos were initially exposed to PBZ at 72 or 96 hpf (Figure 3B). At these later stages (72 and 96 hpf), the neural crest cells have already migrated to the head skeletal locations and undergone differentiation into chondrocytes, which further mineralize to form bones. Our results therefore suggest that PBZ most likely disrupts the neural crest cell population and its migration. Moreover, PBZ might influence some factors that participate in neural crest cell differentiation.

### 3.4. Comparative Toxicity of PBZ in Developing Digestive Organs

The digestive system allows an organism to absorb nutrients after it is born. During embryogenesis, the digestive tract and its accessory organs, including the liver, intestine and pancreas, arise from the same endodermal primordium [25,26,27]. In our previous study, we found the development of digestive organs (i.e., intestine pancreas, and liver) was affected by PBZ via activation of AhR signaling [15]. Therefore, we next sought to determine the timeframe of PBZ toxicity on the development of digestive organs during embryogenesis. In the zebrafish embryos exposed to various concentrations of PBZ with different initial exposure stages, the toxic effects on digestive organs were assessed by whole-mount in situ hybridization with digoxigenin-labeled *ifabp* (intestine), *lfabp* (liver) and *insulin* (pancreas-specific marker gene) antisense riboprobes. The morphology of digestive organs was categorized as ‘normal’, ‘mild’ defects or ‘severe’ defects based on the expression patterns of marker genes (Figure 4A, Figure 5A, and Figure 6A). The intestine is the main part of the digestive tract that absorbs nutrients for cell survival and function. In zebrafish initially exposed to PBZ at 24 and 36 hpf, more than 70% of embryos exhibited mild or severe intestine hypoplasia in the 0.34 and 3.4 μM groups, and 100% of embryos in the 17 μM-treated group showed hypoplasia. The intestine hypoplasia phenotype was significantly less frequent when embryos were exposed to PBZ beginning at 60 hpf, and only a small percentage (4.1 ± 0.3%) of embryos with intestine defects was observed in embryos treated with 17 μM beginning at 96 hpf (Figure 4B).

Liver is a major accessory organ in the digestive system that has many functions, including biosynthesis, biometabolism, immunological function, and detoxification. The development of liver was examined by in situ hybridization staining for the liver-specific marker, *lfabp*. Embryos were exposed to a range of PBZ concentrations beginning at different embryonic stages, and the *lfabp*-stained livers were categorized as ‘normal’, ‘mild’ or ‘severe’ defects (Figure 5A). All embryos initially exposed to all doses of PBZ at 24 hpf showed liver-development defects, and severe defects were dose-dependently increased. Notably, less than 22% of embryos presented severe defects in liver when embryos were exposed to PBZ beginning at later stages (after 48 hpf) even in high concentrations of PBZ (3.4 and 17 μM).

In contrast with the 24 hpf treatment-stage cluster, the developing liver was less sensitive to PBZ when initial exposure occurred at 36 hpf; about 60% of embryos showed mild defects, and 2% of embryos had severe defects in 0.34 μM. A similar phenomenon was observed in 3.4 and 17 μM-treated embryos. Embryos initially exposed to PBZ at 24 hpf all had liver defects (0.34 μM: 38.3% severe, 61.7% mild; 3.4 μM: 57.6% severe, 42.4% mild; 17 μM: 80.3% severe, 19.7% mild).

We also assessed the toxic effect of PBZ on pancreas (Figure 6). Similar to the developing intestine and liver, initial exposure of embryos to PBZ at earlier stages (24 and 36 hpf) led to more frequent and more severe pancreas defects. Among zebrafish embryos initially exposed to PBZ at 24 hpf, 39.3%, 57.8% and 80.1% of embryos exhibited severe pancreas developmental hypoplasia in 0.34, 3.4, and 17 μM PBZ-treated groups, respectively (Figure 6B). Compared to embryos with exposure beginning at 36 hpf, the percentage of embryos with severe defects was significantly higher when embryos were exposed from 24 hpf. This was even true for lower concentrations of PBZ (0.34 and 3.4 μM). Thus, our data suggest that 24–36 hpf is a critical time period for pancreas development. Unlike intestine and liver, less than 35% embryos presented mild defects in the pancreas when treated with 17 μM PBZ. No defects were found with lower concentrations or in embryos with exposures started after 60 hpf. Together, our results indicate that the development of liver and pancreas was not substantially impaired when PBZ exposure began at 96 hpf. In contrast, 10–30% of embryos presented mild intestinal defects at this exposure timing.

## 4. Discussion

PBZ has been used in commercial applications for more than 30 years since it was first registered with the U.S. EPA (1985). According to the U.S. EPA, the acute oral LD50 for mallard ducks is >7900 mg/kg, while the LC50 (96 h) for rainbow trout is 27.8 mg/L, and for Daphnia magna, the LC50 (48 h) is 33.2 mg/L. These values indicate that the acute toxicity of PBZ is low compared to organochlorine and organophosphate pesticides [28], so the potential biological toxicity of PBZ has been ignored for many years. Recently, however, the long-term use, high stability, and high environmental mobility have caused researchers to begin considering the potential ecological risks posed by residual and accumulated PBZ in soil and water. Many lines of evidence indicate that the global use of PBZ has led to significant environmental contamination and residue in food, suggesting it has the potential to present a risk to the health of humans and other organisms.

In recent years, our research group has focused on evaluating the embryonic toxicity of PBZ using a zebrafish model system. Our previous studies demonstrated that embryonic exposure to PBZ can result in developmental hypoplasia of head skeleton, eye and components of the digestive system. During embryogenesis, organs in the digestive system, including intestine, liver and pancreas, are formed from the endoderm, which arises during the process of gastrulation. To explore the different toxic effects of PBZ on these organs, we exposed zebrafish embryos to various concentrations beginning at different embryonic stages. We first examined the survival rate and heart development, finding dramatically decreased survival (around 16%) and high rates of heart developmental hypoplasia when embryos were exposed to 17 μM PBZ at 24 or 36 hpf. However, the survival rates were not so dramatically decreased when exposures began after 48 hpf. This result indicated that the early-stage embryos are more sensitive to PBZ. During zebrafish embryogenesis, cardiac progenitors are first observed at the lateral margin at 5 hpf and bilaterally migrate to the anterior lateral plate mesoderm by 15 hpf. At 22 hpf, the cardiac progenitors and developing endocardial cells fuse to form the cardiac disk and begin regular contractions. From 24 to 36 hpf, the cardiac disk elongates into a linear heart tube and leftward migration begins to establish looping (ventricle on the right and the atrium on the left). Thus, by 48 hpf, the two-chambered heart has formed [29]. Our results indicate that the fish are especially sensitive to PBZ prior to heart tube formation and during the looping process.

Vertebrate head skeleton is derived from neural crest cells, a vertebrate-specific cell population that arises from the crest of neural tube at 12 hpf [30], which ventrally migrates to the pharyngeal arch primordia before 36 hpf. From 36 to 48 hpf, the migrated neural crest cells activate various transcription factors, such as sox9, fgfs, and endothelin-1, which causes their differentiation into chondrocytes [31,32,33]. Our results indicate that exposure of 17 μM PBZ to zebrafish embryos beginning at 24, 36 or 48 hpf causes all embryos to exhibit mild or severe defects in the head skeleton. In our previous study, we found that exposure of 17 μM PBZ to zebrafish embryos from 2 hpf diminishes the number of neural crest cells at 36 hpf and severely affects head skeleton morphology [13]. Combined with the observations from our previous study, our current data suggest that the embryonic toxicity of PBZ on head skeletal development is not only limited to effects on the neural crest population, but PBZ also interferes with differentiation of migrated neural crest cells.

The digestive system consists of the digestive tract and several associated organs, including liver, gallbladder and pancreas, which are all derived from endoderm. The zebrafish digestive organs have been well characterized by molecular, genetic and morphological studies conducted since 1996. At 18 hpf, the pharynx, pancreas and gut progenitors can be identified by staining with tissue-specific marker genes. The foregut structure forms at 21 hpf, and hindgut formation coincident with liver progenitor expansion occurs at 26 hpf. As development progresses, the foregut and hindgut continue to extend (forward and backward), fusing together at 56 hpf. At the same stage, the pancreas bud can be observed within the gut [25,26,27,34]. Our previous study revealed that PBZ exposure causes embryonic liver, pancreas and intestine hypoplasia [15]. Here, we further examined the toxic effects on digestive organs at different exposure timings. Our results revealed that the embryos with initial exposure to PBZ at early stages exhibited higher sensitivity with regard to development of digestive organs (liver, pancreas and intestine). Liver development was highly sensitivity to PBZ when embryos were exposed to PBZ beginning at 24 hpf; at this timing, 100% of exposed embryos presented liver defects (mild or severe). Among embryos exposed to PBZ beginning at 24 hpf, 100% of those exposed to 3.4 or 17 μM PBZ exhibited pancreas defects, and 100% of those exposed to 17 μM PBZ had intestinal defects. Surprisingly, the development of pancreas and liver was more sensitive to PBZ than intestine when embryos were exposed beginning at 24 hpf. On the other hand, only the high concentrations of PBZ (3.4 or 17 μM) could cause liver or pancreas defects in small percentages of embryos if exposure began at 60 hpf. However, when embryos were exposed to PBZ at 60 hpf, the development of intestine was still sensitive to PBZ and only exhibited less sensitivity to PBZ when exposure began at 96 hpf.

In conclusion, or study demonstrates the comparative toxic effects of PBZ exposure timing on survival, and development of heart, head skeleton, liver, pancreas, and intestine. Zebrafish embryos showed higher sensitivity to PBZ when exposure began at earlier stages. Though the digestive organs (liver, pancreas and intestine) arise from the same endodermal origin, each tissue presents a different sensitivity profile to PBZ based on exposure timing. Because our results systematically dissect the effects of different PBZ exposure timings, the information may serve as a valuable reference for further studies on the toxic mechanisms of PBZ in developing organs.

## Figures and Tables

**Figure 1 toxics-07-00062-f001:**
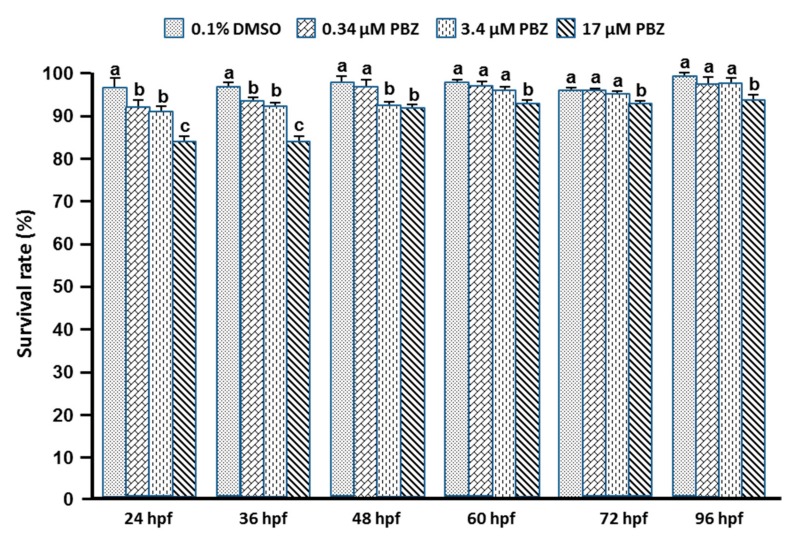
Comparison of PBZ effects on survival rate after exposure of zebrafish embryos beginning at different stages. Survival rates at 5 dpf were calculated for embryos exposed to 0.1% DMSO (control), and 0.34, 3.4, and 17 μM PBZ. Exposures began at 24, 36, 48, 60, 72 or 96 hpf. Data represent the mean (±standard deviation, SD) of three independent experiments; 80 embryos were assessed in each treatment (*N* = 3. *n* = 240 embryos). Bars not sharing a common letter are significantly different. Data were compared by ANOVA followed by Fisher’s least significance difference test (*p* < 0.05).

**Figure 2 toxics-07-00062-f002:**
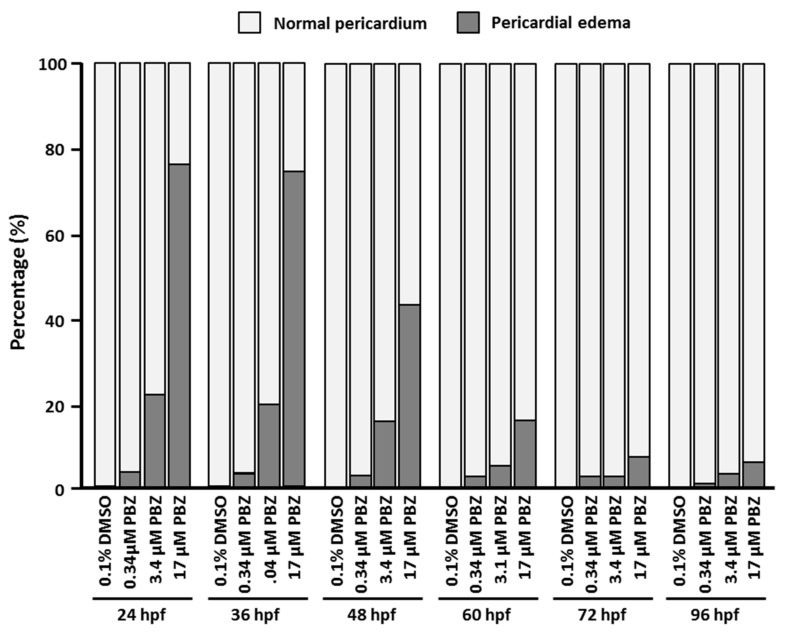
Pericardial edema is induced by PBZ in zebrafish embryos exposed at early stages. Zebrafish embryos were exposed to a range of PBZ concentrations (0.34, 3.4, and 17 μM) beginning at different stages (24, 36, 48, 60, 72, and 96 hpf). The pericardiac phenotype was observed at 5 dpf, and ratios of affected individuals were calculated. Data represent the mean of three independent experiments; 30 embryos were assessed in each treatment (*N* = 3. *n* = 90 embryos).

**Figure 3 toxics-07-00062-f003:**
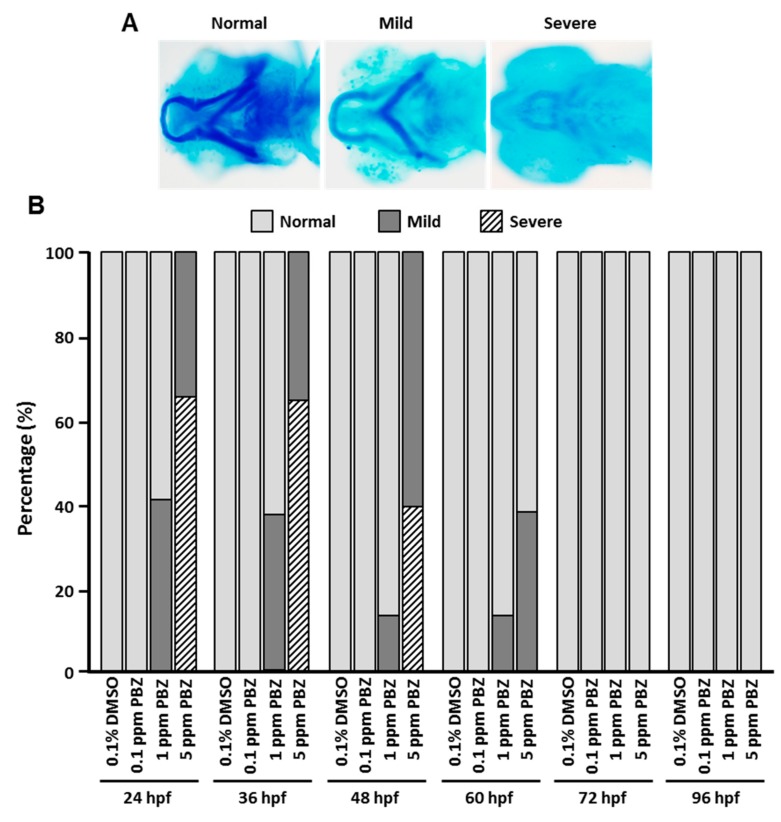
Pharyngeal arch development is affected according to the initial PBZ exposure stage in zebrafish embryos. Zebrafish embryos were exposed to a range of PBZ concentrations (0.34, 3.4, and 17 μM) beginning at different stages (24, 36, 48, 60, 72, and 96 hpf). The head skeleton of 5 dpf fish was analyzed by Alcian blue staining. Pharyngeal arch phenotypes were categorized as (**A**) ‘Normal’, ‘Mild’ defects or ‘Severe’ defects. Images were taken from the dorsal view. (**B**) Frequency of pharyngeal arch phenotypes was compared in zebrafish embryos treated with different concentrations of PBZ at different exposure stages. Data represent the mean of three independent experiments; 30 embryos were assessed in each treatment (*N* = 3. *n* = 90 embryos).

**Figure 4 toxics-07-00062-f004:**
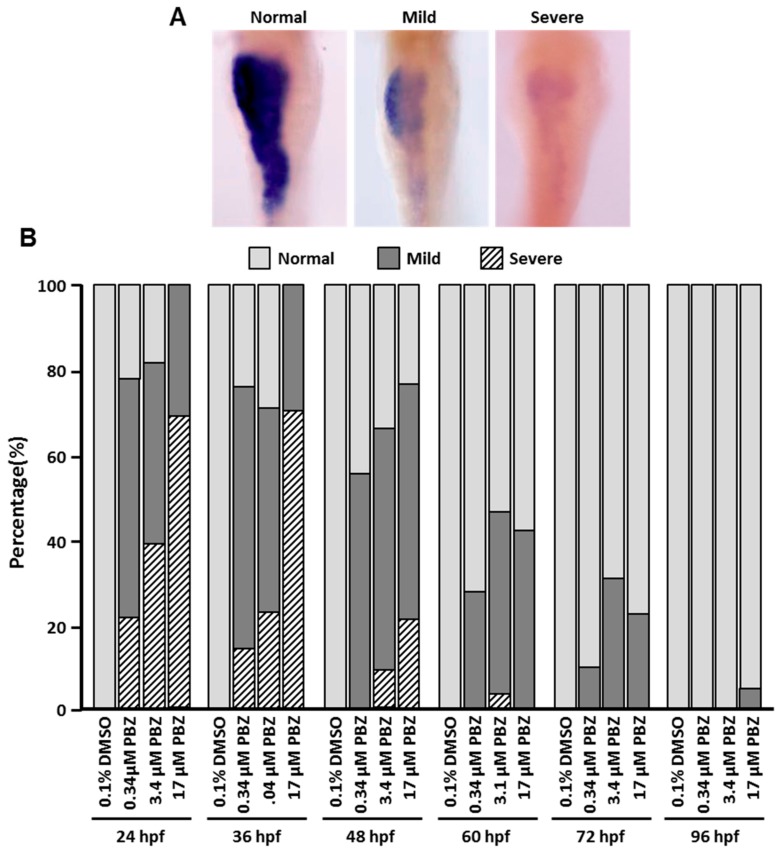
Intestine defects are more severe when exposure to PBZ begins at early stages. Zebrafish embryos were exposed to a range of PBZ concentrations (0.34, 3.4, and 17 μM) beginning at different stages (24, 36, 48, 60, 72, and 96 hpf). Embryos were collected at 5 dpf and analyzed by whole-mount in situ hybridization with an *ifapb* antisense riboprobe; *ifapb* is specifically expressed in intestine. The intestinal phenotype was categorized as (**A**) ‘Normal’, ‘Mild’ defects or ‘Severe’ defects. Images were taken from ventral view. (**B**) Frequency of intestine phenotypes was compared in zebrafish embryos treated with different concentrations of PBZ at different exposure stages. Data represent the mean of three independent experiments; 30 embryos were assessed in each treatment (*N* = 3. *n* = 90 embryos).

**Figure 5 toxics-07-00062-f005:**
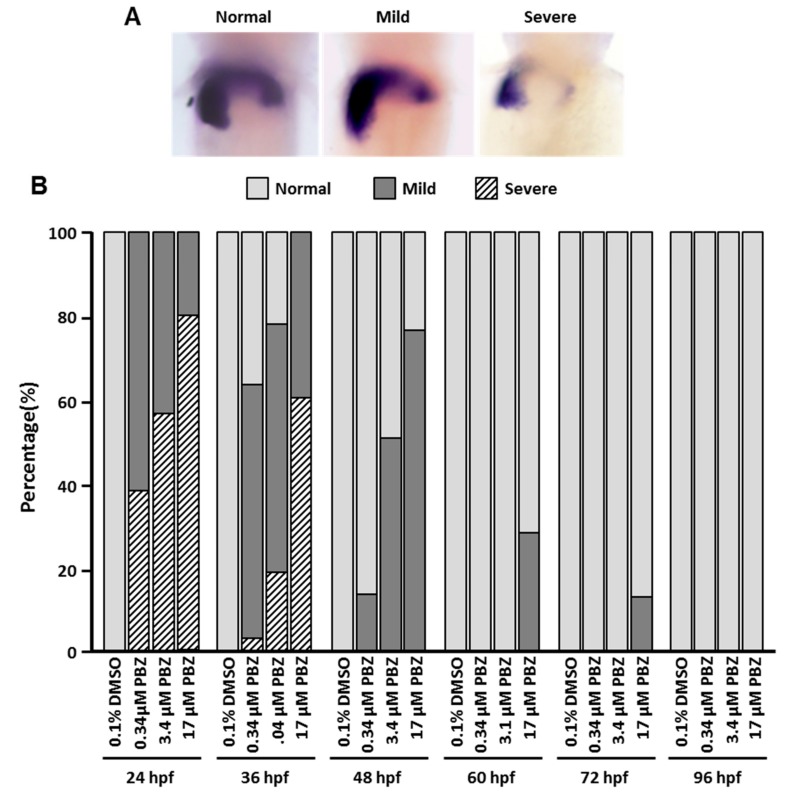
Liver defects are more severe when exposure to PBZ begins at early stages. Zebrafish embryos were exposed to a range of PBZ concentrations (0.34, 3.4, and 17 μM) beginning at different stages (24, 36, 48, 60, 72, and 96 hpf). Embryos were collected at 5 dpf and analyzed by whole-mount in situ hybridization with an *lfpb* antisense riboprobe; *lfpb* is specifically expressed in liver. The liver phenotype was categorized as (**A**) ‘Normal’, ‘Mild’ or ‘Severe’ defects. Images were taken from ventral view. (**B**) Frequency of liver phenotypes was compared in zebrafish embryos treated with different concentrations of PBZ at different exposure stages. Data represent the mean of three independent experiments; 30 embryos were assessed in each treatment (*N* = 3. *n* = 90 embryos).

**Figure 6 toxics-07-00062-f006:**
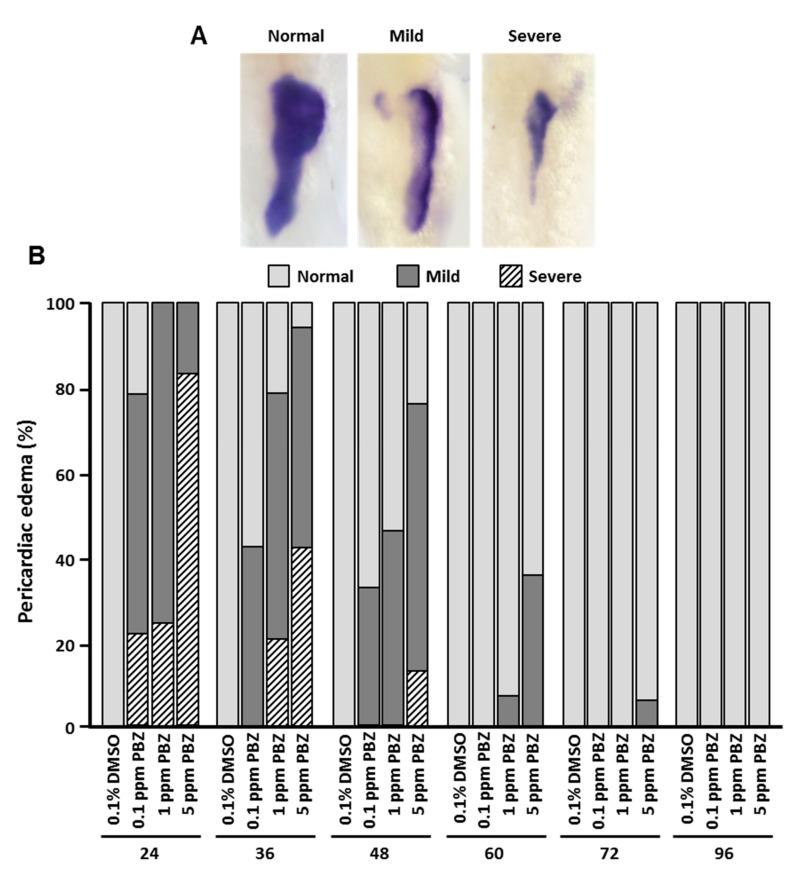
Pancreas defects are more severe when exposure to PBZ begins at early stages. Zebrafish embryos were exposed to a range of PBZ concentrations (0.34, 3.4, and 17 μM) beginning at different stages (24, 36, 48, 60, 72, and 96 hpf). Embryos were collected at 5 dpf and analyzed by whole-mount in situ hybridization with an *insulin* antisense riboprobe; *insulin* is specifically expressed in pancreas. The intestinal phenotype was categorized as (**A**) ‘Normal’, ‘Mild’ or ‘Severe’ defects. Images were taken from dorsal view. (**B**) Frequency of pancreas phenotypes was compared in zebrafish embryos treated with different concentrations of PBZ at different exposure stages. Data represent the mean of three independent experiments; 30 embryos were assessed in each treatment (*N* = 3. *n* = 90 embryos).

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
