# Peer review of "Toxic Effects of Paclobutrazol on Developing Organs at Different Exposure Times in Zebrafish"

_toxics, 2019, doi:10.3390/toxics7040062_

Round 1

Reviewer 1 Report

The manuscript by Wang et al., presents a study in which they investigated the effect of PBZ on toxicity and development of Zebra fish embryos. The toxicity effect was not big as a maximum of 15% death was achieved with the highest concentration and earliest developmental l stage, the authors may change the Y axis scale to a conventional 0 to 100% and they can present a zoom in to highlight the change. However, the effect on the embryonal development was dramatic.

The results are clearly described and presented. I have only few comments

Figure1: The authors should indicate the statistical P value and include stars for each value <0.05, <0.01 or <0.001 (unless they are the same) The authors should indicate the number of embryos used (n) in each figure legend Title of the Y axis should be pericardial edema (%) Figures 2,3,4,5,6: The statistical analyses are mentioned in figure legend but are missing from the figure, what does that mean? The difference is not statistically significant or the authors forgot to include them Figures 3,4,5,6 Y axis title is the same for all figures “pericardiac edema”. The authors may change the title to reflect the specific organ they are analyzing

Author Response

Response:

Thanks for suggestions from reviewer 1.

I have changed the Y axis scale to a conventional 0 to 100%. I have added the number of independent experiments (N) performed and the number of embryos (n) used in the figure legends of Figure 2, 3, 4, 5 and 6. In Figure1, the statistic difference was shown with different letters (the p value is <0.05), and the description is shown in the figure legend. I have corrected the title of the Y axis in Figure 3, 4, 5 and 6. Sorry for the mistakes with the wrong labeling.

Reviewer 2 Report

Authors present results of their studies on toxic effects of PBZ on developing organs.

Please give full name of PBZ in title – it is not so long to abbreviate it; please add also name of model organism You used. Abstract is non-informative, please describe methods You used and main findings of this research. Introduction is properly constructed and not so extensive.

Line 121: what is this value 300,000 ppm? Please use “.” And give mM levels in all places instead of ppm. This methodology is generally not sufficiently detailed.

Results are sound and properly structured, I am impressed with their quality. am impressed with their quality.

In concluding part please add some sentences of future development trends in this field of expertise, what other compounds will You study?

Author Response

Response:

Thanks for the suggestions from reviewer 2.

The title has been changed to “Toxic Effects of Paclobutrazol PBZ on Developing Organs at Different Exposure Times in Zebrafish.” I have converted the concentration of PBZ from 0.1,1 and 5 ppm to 0.34, 3.4 and 17μM, respectively. All the concentrations of PBZ are shown in molarity in the manuscript.